# Toughness Properties of a 50-Year-Old Pipeline Material

**Darko Bajić** 

Faculty of Mechanical Engineering, University of Montenegro, Cetinjski put bb, 81000 Podgorica, Montenegro; darko@ucg.ac.me

**Abstract:** To ensure the reliable functioning of the main pipeline for water transport in hydropower, which has been operational for 50 years, a detailed analysis of the pipeline's material condition is necessary. The pipeline material undergoes changing exploitation and working conditions, including extreme load phases such as periodic intensive emptying and filling with water for inspection. In this paper, we analyse the material resistance of NIOVAL47 steel using the method for determining R-curves for three states: Normalised state (2 samples), aged state (1 sample) which underwent 10% cold deformation and was heated for 30 min at +250 °C, and deformed state (1 sample) which underwent 10% cold deformation. The test results indicate that analysing the material's aged and deformed states is crucial for obtaining a reliable picture of the structural integrity of the NO2500 mm pipeline, which has been in operation for five decades. The study should demonstrate the current state and level of reliability of the pipeline in order to ensure the sustainability of the energy facility. Additionally, these tests provide a realistic picture of the necessity of introducing an online monitoring system for the pipeline.

**Keywords:** pipes; material state; mechanical properties; R-curve; fracture toughness; semi-elliptical crack; stress intensity factor



## 1. Introduction

The reliability of pipe systems is of vital importance for the functioning of electricity-producing power plants [1–5]. Periodic inspections which use NDT (Non-Destructive Testing) methods control the state of critical places [6–8]. These places are welded joints, arcs, reductions, and places where pipe fittings are installed.

The analysed structure is a pipeline ND2500 × 32 mm with an internal working pressure of 50.9 bar. It is an energy facility within the HPP Perućica, Montenegro. The pipeline has been in operation and has been exploited for 50 years.

The parameter that defines the reliability of the structure is the critical value of fracture toughness [9–12]. Determination of one of the parameters ($K_I$—stress intensity factor, J—integral or critical crack opening, and CTOD (Crack Tip Opening Displacement)) is preceded by defining the mechanical tensile characteristics of NIOVAL47 steel. The experiment must be performed in accordance with the ASTM 1820 standard [13].

## 2. Materials and Methods

The material used for the construction of the pipe is NIOVAL47 steel, manufactured by SIJ—Slovenian Steel Group (ex-Steelworks Jesenice, Slovenia).

Three states of NIOVAL47 steel were analysed:

- State A: The normalised state of the material before its deformation and pipe making (undeformed sheets).
- State B: The aged and deformed state, which implies 10% cold deformation and heated for 30 min at +250 °C.
- State C: The deformed state of the material which is in the range of 10% cold deformation.

The mechanical properties of this material are given in Table 1.

**Table 1.** Mean values of the tensile characteristics of the material NIOVAL47 at room temperature.

| Material | $E$, GPa | $R_{p0.2}$, MPa | $R_m$, MPa | $E_m$, % | $E_f$, % | $\sigma_0$, MPa | $N$ |
|---|---|---|---|---|---|---|---|
| State A | 196 | 442 | 610 | 13.7 | 27 | 27 | 27 |
| State B | 193 | 647 | 726 | 4.7 | 12 | 12 | 12 |
| State C | 192 | 627 | 684 | 4.5 | 11 | 11 | 11 |

## 3. Results Analysis and Discussion

### 3.1. Tensile Testing

Tensile test specimens were obtained from rolled sheets of sheet metal to determine the mechanical properties of the material perpendicular to the rolling direction. This direction is typically less favourable in terms of mechanical properties. The test was performed on a sample that represents the least favourable direction. Additionally, the sampling method was chosen to characterise the properties of the material when the pipeline opens (breaks) longitudinally.

The test specimens (Figure 1) were taken from 20 mm thick sheet metal plates (the basic delivered state of the material) and 18 mm thick sheet metal plates (in the deformed and aged state of the material). The tensile test was conducted according to the standard that specifies a test specimen needs to have a 5 mm diameter.

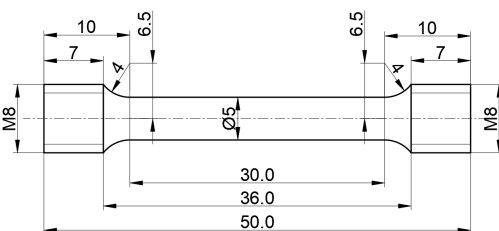

**Figure 1.** A tension test specimen with a 5 mm diameter.

The tests were performed at the temperature of +22 °C, using the servo-hydraulic testing machine INSTRON 1255 with the 250 kN capacity, with a constant speed of the cylinder piston at 1 mm/min. Based on the data for the limit of proportionality $\sigma_0$ and the deformation strengthening exponent $N$ (Table 1), we obtained curves of mean values that characterise the correlation between true stress and true strain $\sigma$–$\varepsilon$. The tension curves are shown in Figure 2. For the numerical calculation, it is necessary to take into account the material input data corresponding to the state of material B (aged and deformed state). Since the material NIOVAL47 was already plastically deformed during the construction of this pipeline in 1974, due to the technical operation of rolling, the mechanical properties of the material in the certificate of the supplier SŽ Jesenice no longer correspond to the actual state of the material of the structure in question. As can be seen (Table 1 and Figure 1), the value of the yield strength of the material in the deformed and aged state (state B) is significantly higher than the value of the yield point of the material in the delivered normalised state (state A), which has traditionally been taken as the initial state for numerical calculation.

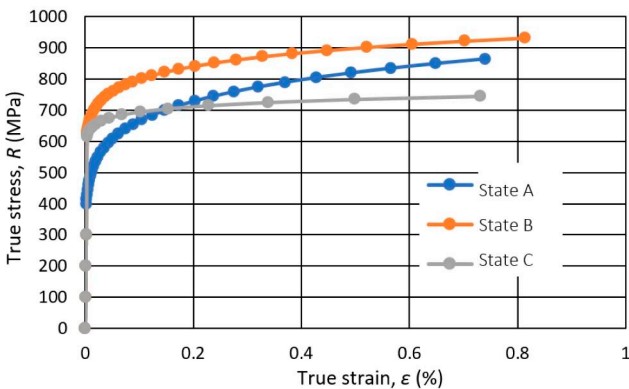

**Figure 2.** Tensile curves for the three material states.

### 3.2. Examination of Fracture Mechanics

Experimental determination of fracture toughness was performed using compact tension specimens (CT) in accordance with the ASTM 1820 standard. Two test specimens were tested against the delivered state (state A), one for the aged and deformed state (state B) and one for the deformed state, respectively (state C). The test specimens were sampled from sheet metal plates with a thickness of 20 mm (as the new delivered state of the material) and from metal sheet plates with a thickness of 18 mm (in the deformed and aged state of the material). The required 10% deformation was achieved by cold rolling from a thickness of 20 mm to 18 mm. As all plates were uneven, and most of them were deformed (bending) after the thickness reduction from 20 mm to 18 mm, it was necessary to straighten them. With this technological procedure, their thickness was reduced by another 2 mm, i.e., by another 10%. The test specimens are made of 200 × 500 mm plates (length × width) with a mechanical notch in the direction of rolling. In this way, states are created for determining the fracture toughness of the material, which is relevant to the opening of longitudinal cracks. Furthermore, the fracture toughness values for the sampled test specimens with a mechanical notch perpendicular to the rolling direction are higher than the equivalent ones measured in the sheet rolling direction [14]. In this way, the state of conservatism is met, because the measured fracture toughness values with a notch perpendicular to the rolling direction will be higher than the values with a notch in the rolling direction. The nominal dimensions of the test specimen are shown in Figure 3. Before starting the fracture toughness test, the test specimen was fatigued in order to achieve the required crack length with the smallest possible plastic zone at the microstructural level [13,15,16]. Material fatiguing was performed on the servo-hydraulic testing machine INSTRON 8500+, as shown in Figure 4. In accordance with the standard, the length of the fatigue crack must be greater than 5% of the entire length of the crack $a_0$, i.e., greater than 1.5 mm.

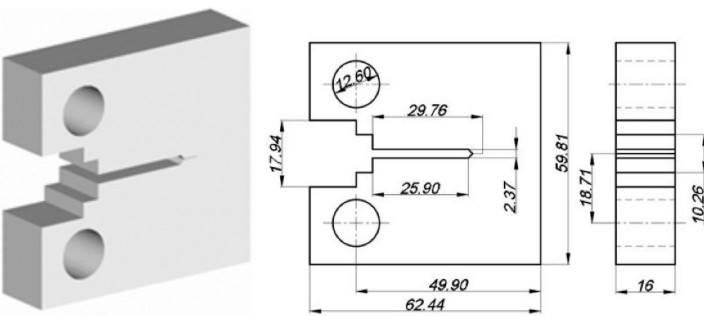

**Figure 3.** Shape and dimensions of compact tension specimens (CT specimen) for determining fracture toughness (CT specimen thickness $B$ = 16 mm (states B and C) and $B$ = 18 mm (state A), CT specimen width $W$ = 62.44 mm).

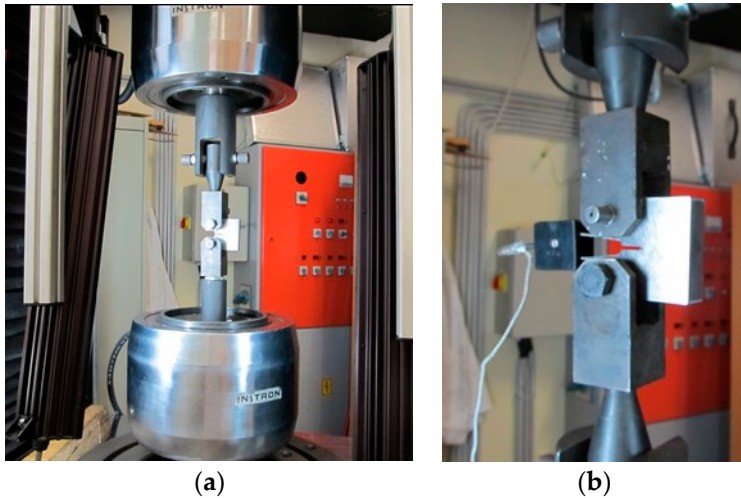

**Figure 4.** Fatigue (**a**) and testing (**b**) of the compact tension specimen on the testing machine INSTRON 8500+.

An additional requirement of the standard is that the length of the crack meets the state, i.e., that the ratio of the length of the crack *a* and the width of the compact tension specimen *W* is within $0.45 \leq a/W \leq 0.6$. Determination of fracture toughness was carried out at a constant displacement speed of $v = 1$ mm/min. The test was controlled using the testing machine INSTRON 8500+ management console and the "INSTRON View-maker" software. During the experiment, we monitored the dependence of force—displacement (*F*—$\Delta$) on the point of impact of the force as well as force—Crack Mouth Opening Displacement (*F*—CMOD). Figure 5 shows the *F*—CMOD curves for four compact tension specimens (3 material states). The diagram shows that, with the compact tension specimens in state A, there is a blunting of the crack and plastic deformation with the force increasing to its maximum value only for CMOD values between 3 and 4 mm. In the case of compact tension specimens for the material in state B or state C, the maximum force values were reached for a CMOD value of up to 1 mm, followed by a stable crack growth with a drop in the force value. They quickly reached maximum force and the decreasing curve shows that the materials in states B and C have a lower resistance to stable crack growth, even though these materials do not experience a brittle fracture [17].

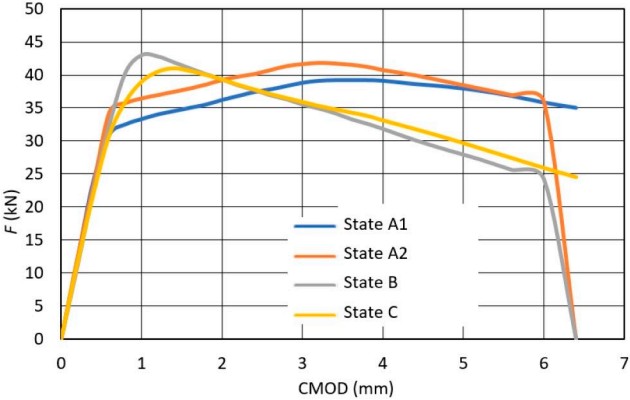

**Figure 5.** Curves force—Crack Mouth Opening Displacement (*F*—CMOD).

After the completion of the first phase of the fatigue test, the specimens were again subjected to dynamic loading to determine the length of the stable crack growth. In the case of broken specimens, the initial and final average lengths of the crack were measured. These data are necessary to determine the fracture mechanics parameters pertaining to the test specimens. The normalisation technique [18–20] was used to determine the fracture

toughness. For each tested specimen, J-R curves of material resistance with the corresponding final length of the stable crack growth were formed. The test results were evaluated in accordance with the ASTM E-1820-05a standard [21] and are shown graphically in Figure 6 for all four tested specimens. The relevant values are given in Table 2.

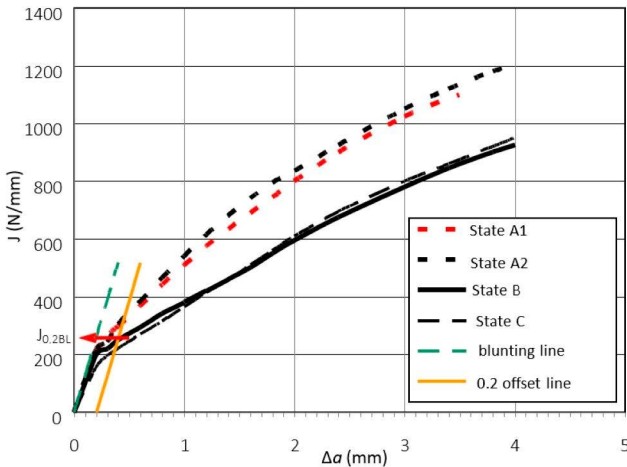

**Figure 6.** J-*R* resistance curves for four test specimens.

**Table 2.** Test results for determining fracture mechanics parameters.

| Parameter | Label | A1 | A2 | B | C |
|---|---|---|---|---|---|
| Initial crack length | $a_0$, mm | 25.052 | 28.089 | 27.797 | 29.766 |
| The ratio of the length of the crack to the width of the compact tension specimen | $a_0/W$ | 0.561 | 0.562 | 0.556 | 0.595 |
| Stable increase in crack length | $\Delta a_{stab}$, mm | 3.495 | 4.020 | 4.412 | 4.525 |
| Crack opening 0.2BL * | $J_{0.2,BL}$, N/mm | 298 | 305 | 250 | 219 |
| Crack length increment at 0.2BL | $\Delta a_{0.2BL}$, mm | 0.570 | 0.540 | 0.307 | 0.336 |
| Maximum force | $F_{max}$, kN | 39.1 | 41.9 | 42.9 | 41.2 |
| Value at $F_{max}$ | $J_m$, N/mm | 635 | 740 | 340 | 315 |
| Fracture toughness of material PSS ** | $K_{I,mat}$, MPa·m$^{0.5}$ | 242 | 244 | 220 | 206 |
| Fracture toughness of material PSN ** | $K_{I,mat}$, MPa·m$^{0.5}$ | 253 | 256 | 230 | 216 |

*—BL—Blunting Line. **—fracture toughness of the material at $J_{0.2,BL}$, for PSS—plane state of stress, PSN—plane state of strain.

### 3.3. Calculation of Critical Crack Length

In accordance with the Structural INtegrity Assessment Procedure (SINTAP procedure) [22,23], a calculation was made for the longitudinal and meridian crack on the flat part of the pipeline, in several places in front of branch junction 6A (Figure 7), behind the fork and on the drainage pipeline towards fork 6B. As the numerical calculations using the software for the finite element method KOMIPS [24] showed that the stresses only locally reach the value of 288 MPa, at the junction of the elliptical connecting plate and the distribution pipes of the fork (the main part of the pipeline and drain), then it is appropriate to estimate the critical crack size at the point of the pipeline where the wall thickness decreases. The stresses in the pipe walls at those places correspond to the theoretical stresses due to the internal pressure of the *p* = 50.9 bar. This approach is conservative, but with it, we can achieve greater reliability when calculating the integrity of the structure. This practically means that the critical size (length) of the crack will be at least equal to the calculated value, and may be higher, which is in accordance with the SINTAP procedure approach.

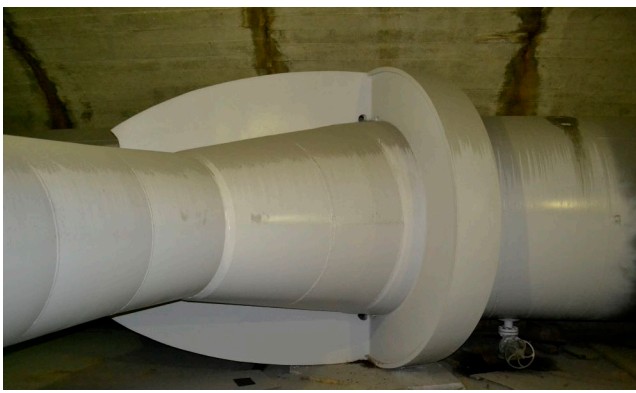

**Figure 7.** The branch junction 6A for which the calculation of the critical crack size is performed.

Individual places on the pipeline for which the hypothetical critical crack length was calculated are shown in Figure 8. These are places of reduction to a smaller pipe thickness, where longitudinal or meridian cracks could occur due to higher stress concentration. The shape of the internal longitudinal semi-elliptical crack is shown schematically in Figure 9a. In addition to the internal longitudinal semi-elliptical crack, it is necessary to calculate the critical crack size for the crack through the pipeline wall (semi-elliptical through a crack). This is an example of the presence of a crack where water could leak rather than destroy the pipeline. Two cases were analysed: The existence of a longitudinal passing crack (Figure 9b) and a meridian internal crack (Figure 9c). In the case of an internal semi-elliptical crack, the change in the length of the crack (2*c*) and the depth (*a*) of the crack in the wall are monitored. In the case of a crack passing through the thickness of the wall, only its length on the surface (2*a*) is monitored.

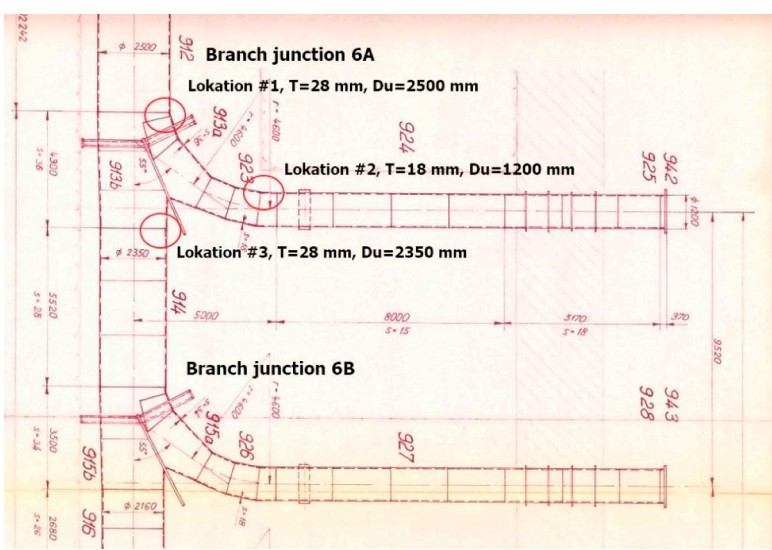

**Figure 8.** Places on the pipeline at branch junction 6A for which the calculation of the critical crack size is performed.

Based on the obtained experimental results, and in accordance with the procedures for assessing the integrity of the structure (SINTAP, EPRI, R6, WES 2805), a calculation can be made in order to determine the critical length of the crack for the adopted load.

Due to the height of the water column, the defined load of the branch junction pipeline is a working internal pressure of 50.9 bar (*p* = 5.09 MPa). For different pipeline diameters and wall thicknesses, different sizes of crack opening stress are obtained in both the longitudinal and meridian directions.

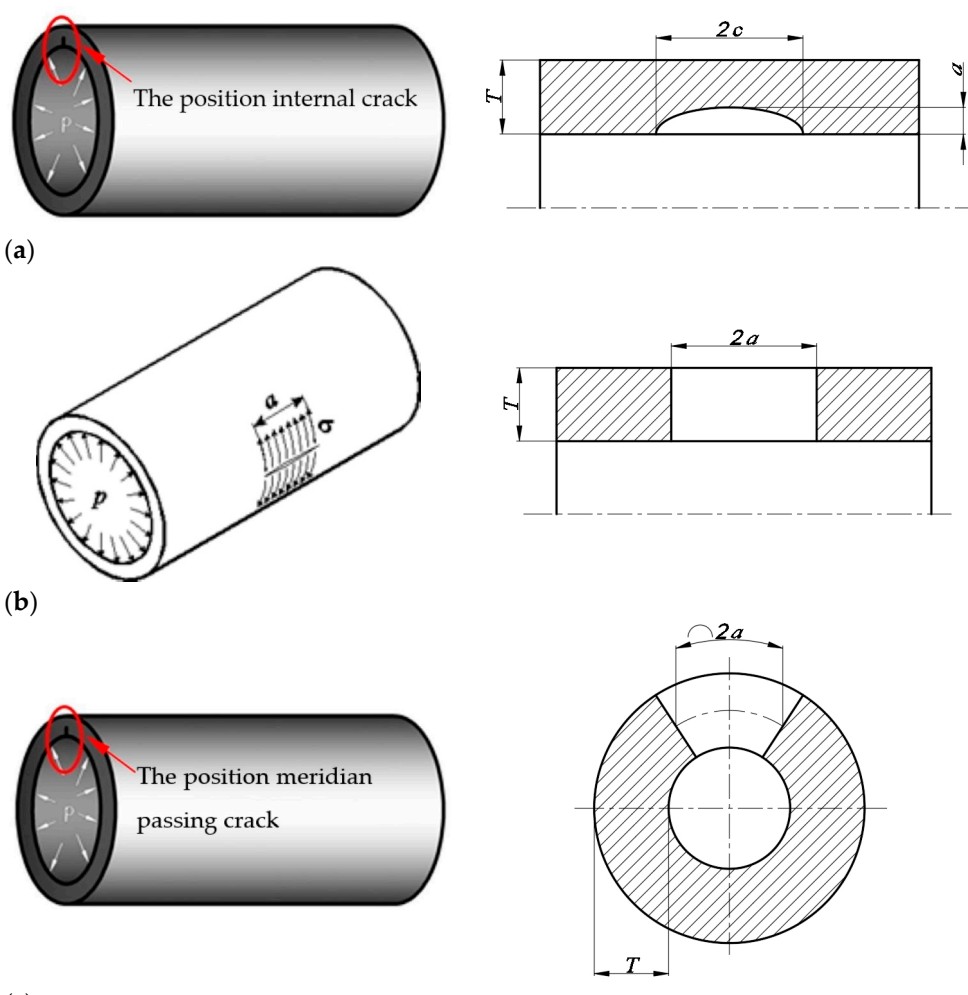

**Figure 9.** Location of the crack: (**a**) Internal longitudinal semi-elliptical crack in the pipeline wall under the effect of internal pressure *p*; (**b**) Longitudinal passing crack through the pipeline wall under the effect of internal pressure *p*; (**c**) Meridian passing crack through the pipeline wall under the influence of internal pressure *p*.

For the longitudinal crack opening, the crack opening stress value ($\sigma_{lon}$) is determined from the following expression:

$$\sigma_{lon} = \frac{p \cdot D_u}{2 \cdot T} \tag{1}$$

For the meridian crack opening, the stress value ($\sigma_{mer}$) of the crack opening is determined from the following expression:

$$\sigma_{mer} = \frac{p \cdot D_u}{4 \cdot T} \tag{2}$$

where *p* bar is internal pressure, $D_u$ mm is the internal pipe diameter, and *T* mm is the pipe wall thickness.

The required mechanical properties for the deformed and aged state of the material (state B) were obtained by testing and are given in Table 1. The relevant value of the fracture toughness at which crack growth occurs for the material in state B is $J_{0.2,BL}$ = 250 N/mm, that is, for the plane state of stress (PSS)—$K_{I,mat}$ = 220 MPa·m$^{0.5}$ or for the plane state of strain (PSN)—$K_{I,mat}$ = 230 MPa·m$^{0.5}$.

Following the original documentation [25–29] and the assumed position and location of the cracks (Figures 8 and 9) on the branch junction pipeline, the values of the pipeline diameter and wall thickness were taken, as indicated in the following tables for each crack.

SINTAP Procedure, Level "1"

The SINTAP procedure is based on two mutually equivalent procedures for the calculation of structural integrity:

- R6, which was developed by British Energy [30,31] and
- ETM, which was developed by HZG (Helmholtz-Zentrum Hereon) [32].

The R6 procedure was used in the calculation, which is based on the FAD (Failure Assessment Diagram) diagram for assessing the critical size of the crack. The FAD concept is based on the FAC (Failure Assessment Curve) curve for assessing the acceptability of a crack-type defect in materials. The diagram is formed based on the normalised load $L_r$ and the error tolerance function $f(L_r)$, (Figure 10). The error tolerance evaluation curve $f(L_r)$ is limited by the $L_{rmax}$ value in the area of plastic collapse.

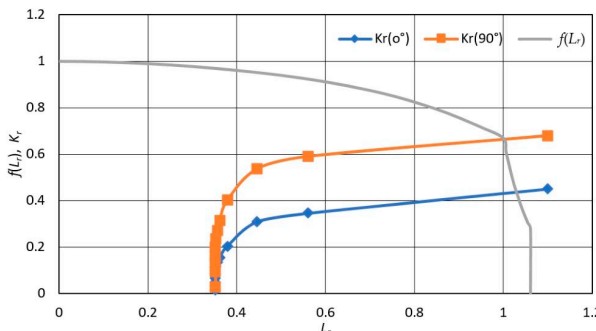

**Figure 10.** FAD diagram for determining the critical crack size.

The process of assessing the integrity of the structure is carried out with constant exposure of the pipe to internal pressure, and with the simulation of crack growth. The procedure is repeated as many times as necessary so that the points in Figure 10 from the area of reliable operation cross the FAC error tolerance curve. Therefore, at the critical dimension and load, the following state [33] will be fulfilled:

$$K_r = f(L_r) \qquad (3)$$

The dimension of the crack at which we found the intersection of the curve of change in crack size and acceptance of the error is the critical dimension of the crack ($a_c$). For each crack length, at the internal pressure of $p$ = 5.09 MPa, it is necessary to determine the normalised value of the stress intensity factor ($K_r$) and the normalised value of the load ($L_r$).

The normalised value of the stress intensity factor $K_r$ is the ratio of the stress intensity factor $K_I(a, \sigma)$ and the measured fracture toughness of the material $K_{mat}$ [33]:

$$K_r = \frac{K_I(a,\sigma)}{K_{mat}} \qquad (4)$$

Using $K_I(a,\sigma)$ or $J_I$, the load level due to pressure ($p$), the position and length of the crack ($a$), as well as the geometric shape of the structural component—the pipeline—are taken into account.

$K_{mat}$ is the fracture toughness parameter of the material, and in the case of brittle behaviour, $K_{mat} = K_{Ic}$.

The calculation of the permissible crack size was made with the measured value of the J—integral $J_{0.2,BL}$ =250 N/mm.

The standard load ($L_r$) is the ratio of the working pressure ($p$) and the pressure at the yield point of the material ($p_Y$), at which the stress will be equal to the yield stress of the material ($\sigma_Y$):

$$L_r = \frac{\sigma}{\sigma_Y} = \frac{p}{p_Y} \qquad (5)$$

The stress σ is equal to the circumferential stress in the pipeline wall and depends on its diameter and wall thickness at a given location [33].

For materials without Lüders plateau (Lüders bands) [34,35] in tensile behaviour, the curve for assessing the acceptability of the FAC error is determined by the following equation [33]:

$$f(L_r) = \left[1 + \tfrac{1}{2}L_r^2\right]^{-1/2} \cdot \left[0.3 + 0.7e^{-0.6L_r^6}\right]$$
$$0 \le L_r \le L_{r\max}$$
(6)

Plastic collapse occurs at a value of $L_r > 1$. As component failure will not occur immediately after reaching $L_r = 1$, it is adopted that the limit engineering value $L_{r\max}$ corresponds to the arithmetic mean of the yield strength ($R_{p0.2}$) and the tensile strength ($R_m$) of the material:

$$L_{r,\max} = \frac{(R_{p0.2} + R_m)}{2 \cdot R_m}$$
(7)

Equations from (3) to (7) are valid regardless of the direction of crack growth or the way and direction the structural component is loaded. This is explained by the fact that both parameters, $K_I(a, \sigma)$ and $L_r$, are determined by the geometric characteristics of the structure and the way it is loaded.

In the SINTAP procedure, already confirmed mathematical and empirical solutions are used for $K_I(a, \sigma)$ and $L_r$, so that for each individual crack configuration, it is stated according to which solutions for $K_I(a, \sigma)$ and $L_r$ we performed the calculation and whether the states are reached.

Longitudinal crack

Figures 11–13 show the diagrams of determining the critical size of the crack (semi-elliptical) at locations #1, #2, and #3 (Figure 8), and for the following ratio: $a/c = 0.1$. Table 3 shows the calculated opening stress and crack geometry on the inner surface for this ratio.

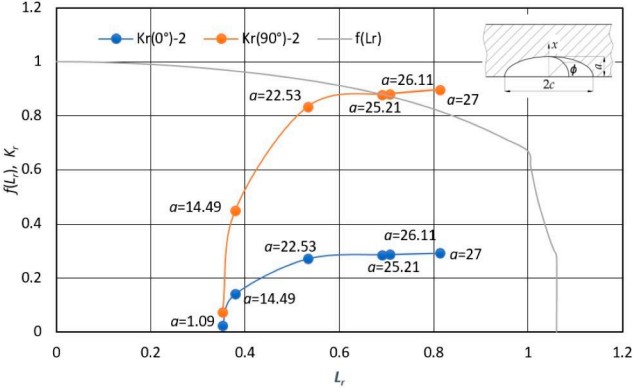

**Figure 11.** Determination of the critical crack size (semi-elliptical) at location #1 for the ratio $a/c = 0.1$.

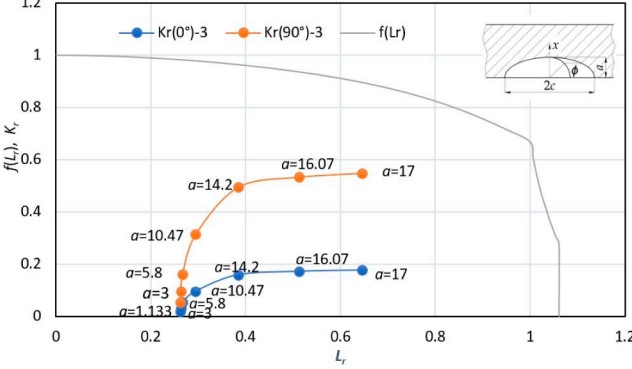

**Figure 12.** Determination of the critical crack size (semi-elliptical) at location #2 for the ratio $a/c = 0.1$.

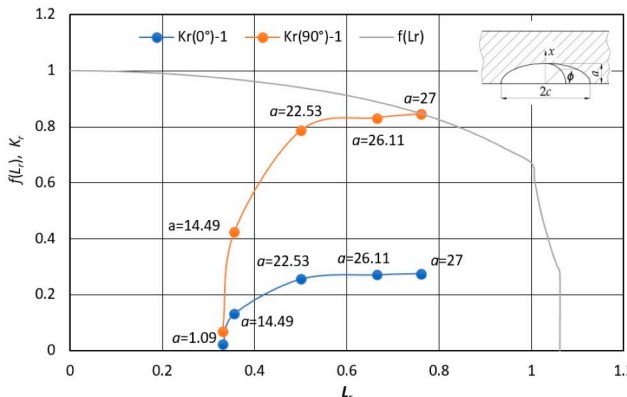

**Figure 13.** Determination of the critical crack size (semi-elliptical) at location #3 for the ratio $a/c$ = 0.1.

**Table 3.** Calculation for the ratio $a/c$ = 0.1 between crack depth $a$ and half of its length on surface $2c$ (Figure 9a).

| Location | $T$, mm | $D_u$, mm | $a/c$ | $\sigma$, MPa | $a_c$, mm | $2c_c$, mm | Comment |
|---|---|---|---|---|---|---|---|
| #1 | 28.0 | 2500 | 0.1 | 227.2 | 25.05 | 500.1 | Fracture before highlighting |
| #2 | 18.0 | 1200 | 0.1 | 169.7 | 17.0 * | 340.0 * | The condition for LBB |
| #3 | 28.0 | 2350 | 0.1 | 213.2 | 26.08 | 521.6 | Fracture before highlighting |

$D_u$—pipe diameter; $T$—wall thickness; $\sigma$—opening stress; $a_c$—critical crack depth; $2c_c$—critical crack length on the inner surface; *—the last value taken for the crack size that is not yet critical.

The stress intensity factor is calculated according to reference [36] with the note that the range of validity is as follows: $0 < T/D < 2$, $0 < a/c < 1$.

From Figures 11–13, it is visible that the crack will reach the critical depth ($a_c$) rather than the critical length ($2c_c$) on the inner surface. The basis for this conclusion is the fact that higher $K_r$ values were calculated for the angle of $\phi = 90°$ than for $0°$. Additionally, the critical depth is reached before the crack passes through the thickness of the wall, which shows that the conditions for leakage of water before breaking (LBB—Leak Before Break) are not met. This indicates it is necessary to do online monitoring for cracks at places #1 and #3.

In Figure 12, it can be seen that all the points for the hypothetical crack depths remain in the diagram within the region of safe operation (below the $f(L_r)$ curve), which indicates the possibility of leakage before failure (LBB). As we continued our work, the calculation was repeated for the example of a passing crack, to determine the critical size of the crack when water leaks from the pipeline.

The limit load is calculated according to reference [37] with the note that the range of validity is $0 < a/T \leq 0.8$, $0.2 \leq a/c \leq 1$.

If the crack depth exceeds $0.8T$, the results for the ultimate load are outside the valid range for the ratio of $a/c$ = 0.1. Therefore, a calculation was made for the ratio of $a/c$ = 0.2 to obtain valid values for the calculation of the critical crack and limit loads (Table 4).

At all three locations, the diagrams in Figures 14–16 show that none of the cracks are critical for failure to occur even though there is only 1 mm left to break through the pipe wall. Since at the last value, at which the calculation is stopped, the character of the crack changes, instead of an internal semi-elliptical (surface) crack, a crack that passes through the pipe wall (through the crack) appears. In the case of a crack that has broken through the pipe wall, leakage occurs and only one geometric parameter of the crack is relevant: Its surface length $2c$. The calculation results are shown in Figures 17–19 and in Table 5.

**Table 4.** Calculation for the ratio *a/c* = 0.2 between the depth of the crack (*a*) and half of its length on the surface (2*c*) (Figure 8a).

| Location | $T$, mm | $D_u$, mm | $a/c$ | $\sigma$, MPa | $a$, mm | $2c$, mm | Comment |
|---|---|---|---|---|---|---|---|
| #1 | 28.0 | 2500 | 0.2 | 227.2 | 27.0 * | 270.0 * | |
| #2 | 18.0 | 1200 | 0.2 | 169.7 | 17.0 * | 170.0 * | The condition for LBB |
| #3 | 28.0 | 2350 | 0.2 | 213.2 | 27.0 * | 270.0 * | |

$D_u$—pipe diameter; $T$—wall thickness; $\sigma$—opening stress; $a$—crack depth; $2c$—crack length on the inner surface; *—the last value taken for the crack size that is not yet critical.

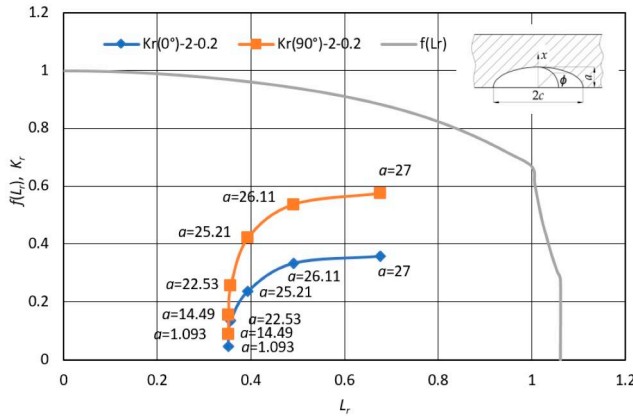

**Figure 14.** Determination of critical crack size (semi-elliptical) at location #1 for ratio $a/c$ = 0.2.

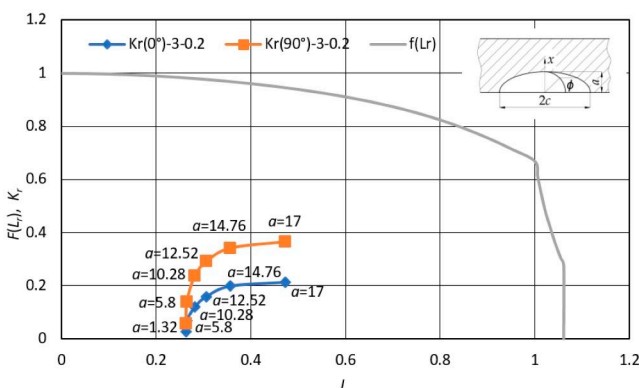

**Figure 15.** Determination of critical crack size (semi-elliptical) at location #2 for ratio $a/c$ = 0.2.

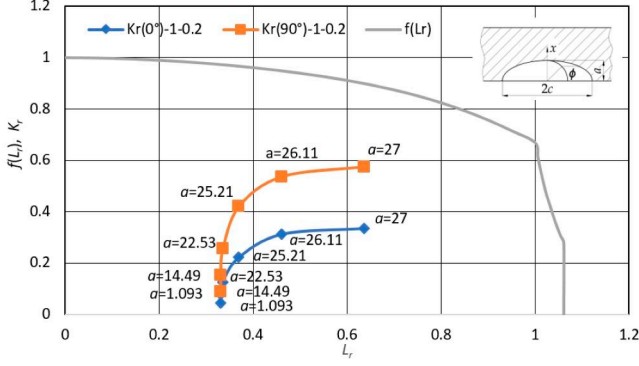

**Figure 16.** Determination of the critical size of the crack (semi-elliptical) at location #3 for ratio $a/c$ = 0.2.

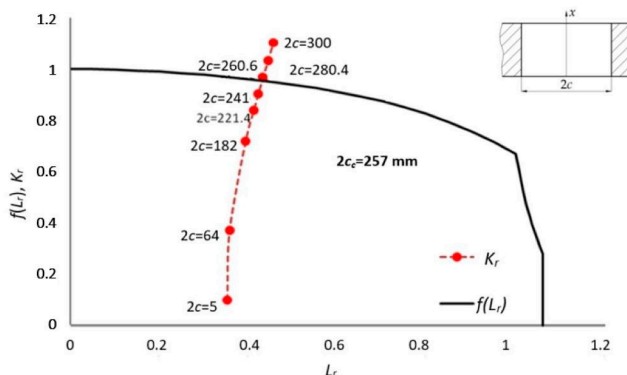

**Figure 17.** Determination of the critical size of the longitudinal passing crack at location #1.

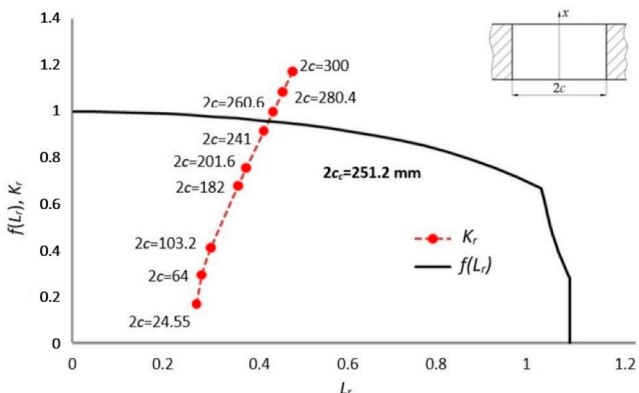

**Figure 18.** Determination of the critical size of the longitudinal passing crack at location #2.

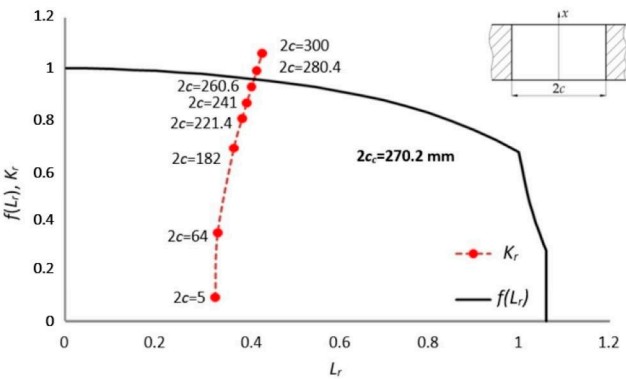

**Figure 19.** Determination of the critical size of the longitudinal passing crack at location #3.

**Table 5.** Calculation for a passing crack that is parallel—longitudinal to the flow of water (Figure 9b).

| Location | $T$, mm | $D_u$, mm | $\sigma$, MPa | $2c_c$, mm | Comment |
|---|---|---|---|---|---|
| #1 | 28.0 | 2500 | 227.2 | 257.0 * | The |
| #2 | 18.0 | 1200 | 169.7 | 251.2 * | condition for |
| #3 | 28.0 | 2350 | 213.2 | 270.2 * | LBB |

$D_u$—pipe diameter; $T$—wall thickness; $\sigma$—opening stress; $2c_c$—critical crack length on the inner surface; *—the last value taken for the crack size that is not yet critical.

Figures 17–19 show the determination of the critical size of a longitudinal passing crack through the pipe wall. Asll determined critical sizes of cracks are of such dimensions that, in the event of their possible formation on a real object, they can be monitored (controlled)

by a system of linear measuring tapes, which would measure the considerable relaxation of the material. In addition, the calculation shows that, in the case of subcritical cracks, water will flow out of the pipeline.

The stress intensity factor $K_I$ was calculated according to the reference [38], and the limit load calculation [39]. "Limit Load Solution" references were used and they give valid results for all shown hypothetical crack dimensions, with a note that the range of validity is $0.01 < T/D < 0.4$, $0.5 < c/T < 25$.

Meridian crack

Since the opening stresses in the case of a meridian crack are two times smaller than the corresponding stresses in the case of a longitudinal crack, we determined the critical size of the crack that passes through the wall (passing crack) and leads to the discharge of water from the pipeline.

Figures 20–22 show the diagrams for determining the critical size of a transient meridian crack at locations #1, #2, and #3 (Figure 8).

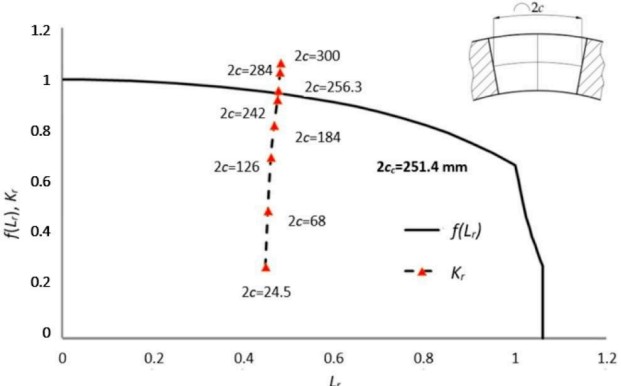

**Figure 20.** Determining the critical size of a transient meridian crack at location #1.

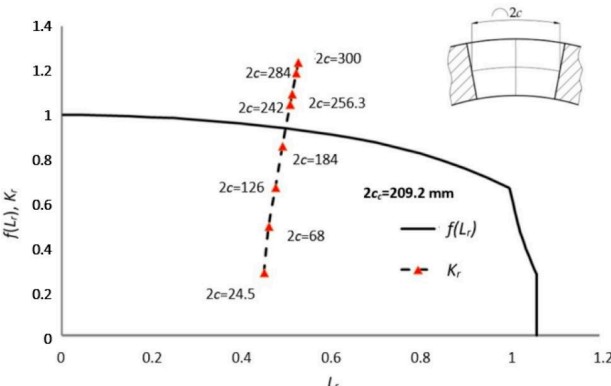

**Figure 21.** Determining the critical size of a transient meridian crack at location #2.

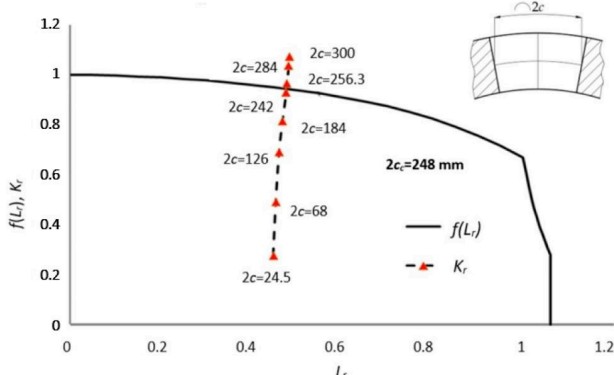

**Figure 22.** Determining the critical size of a transient meridian crack at location #3.

Figures 20–22 show the critical size of a meridian crack that passes through the thickness of the pipe wall. All critical sizes are of significantly high dimensions (Table 6), so any subcritical crack that may have appeared can be registered in a timely manner. At subcritical sizes, water will flow out of the pipeline, but not rupture the pipeline material.

**Table 6.** The calculation for a passing crack that is meridian to the water direction (Figure 9c).

| Location | $T$, mm | $D_u$, mm | $\sigma$, MPa | $2c$, mm | Comment |
|----------|---------|-----------|---------------|----------|---------|
| 1 | 28.0 | 2350 | 106.8 | 248.0 | They are fulfilled conditions for LBB |
| 2 | 28.0 | 2500 | 113.6 | 251.4 | |
| 3 | 18.0 | 1200 | 84.8 | 209.2 | |

$D_u$—pipe diameter; $T$—wall thickness; $\sigma$—opening stress; $2c$—the length of the crack along the inner surface.

The stress intensity factor $K_I$ was calculated according to reference [38] with the validity range of $0.01 < T/D < 0.4$, $0.5 < c/T < 25$. The limit load calculation [40] "Limit Load Solution" is within the scope of validity (thin-walled cylinders). References provide valid results for all hypothetical crack dimensions shown.

## 4. Conclusions

Applying the SINTAP procedure provides answers to the following questions important for the safe operation of pipelines:

1. What is the size of the fatigue crack at which an unstable fracture would occur?

2. Is it possible to observe such a crack with the naked eye, that is, by using the available measuring equipment?

3. Will there be leakage of the medium (water) from the pipeline at a subcritical crack size, which may indicate the need for a timely shutdown of the pipeline system?

Based on the fracture mechanics test results, it can be seen that the fracture toughness values differ depending on the state of the material. The material in the delivered state (state A) shows completely different properties than the material in the aged and deformed state (state B) or the deformed state (state C), both in the case of tensile tests of the test specimens and the tests of fracture mechanics parameters. The fracture toughness values between the deformed and aged states differ slightly, which is why the measured values for the aged and deformed state (state B) can be considered authoritative in the integrity analysis. The fracture toughness of the material is $J_{0.2,BL}$ = 250 N/mm, i.e., $K_{mat}$ = 220–230 MPa·m$^{0.5}$, so it must be taken into account as such when determining the permissible crack size in the pipeline.

**Funding:** This research was funded by the national energy company Elektroprivreda Crne Gore AD Nikšić (Contract No. 20-00-375).

**Institutional Review Board Statement:** Not applicable.

**Informed Consent Statement:** Not applicable.

**Data Availability Statement:** Not applicable.

**Acknowledgments:** The author thanks the support by the national energy company Elektroprivreda Crne Gore AD Nikšić.

**Conflicts of Interest:** The author declare no conflict of interest.

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
