# Peer review of "Toughness Properties of a 50-Year-Old Pipeline Material"

_sustainability, doi:10.3390/su15065143_

Round 1

Reviewer 1 Report

Overall, the authors presented a thorough investigation of a 50-yr old pipeline. This article covers detailed analysis from different aspects, showing that it is necessary to analyze in detail to obtain reliable conclusions about the structural integrity. The data and experimental procedure can be used as future references for people of the same field. There are some minor language issues that need to be modified.

I think this can be accepted after these changes.

1.       Some wording/ writing issues, such as on Page 3 line 79, I am not sure if this is a false expression or a grammar issue. Please check your manuscript thoroughly.

Author Response

Respected,

expresses gratitude to the reviewers for well-founded suggestions that contribute to improving the level of quality of work and do not leave the reader with the possibility of doubts and ambiguities. Thank you!

Reviewer

  1. Some wording/ writing issues, such as on Page 3 line 79, I am not sure if this is a false expression or a grammar issue. Please check your manuscript thoroughly.

Author: Rewiever's remark accepted. Professional proofreading of the manuscript was performed.

Reviewer 2 Report

The manuscript could be accepted with some suggestions for improvement as below:

Title: It is suggested to revise the title of the manuscript to “ Toughness properties of 50-year-old pipeline material”

Abstract:

(1)   Number of sampling should be provided in the abstract

(2)   Significance of the study should be added at the end of the abstract

Introduction

Literature study of the following topic supported with current references should be provided:

(1)   Effect on pipe systems towards the functioning of power plants

(2)   Periodic inspections NDT (Non-Destructive Testing) of different types of pipeline system

(3)   Effect of stress intensity factor, integral or critical crack opening on mechanical properties of pipeline

Results and discussion

Please justify the following discussion with support of recent references:

(1)   yield strength of the material in the deformed and aged state (state B) is significantly  higher than the value of the yield point of the material in the delivered normalized state of the material

(2)   crack length with the smallest possible plastic zone at the microstructural level.

(3)   crack and plastic deformation with force increasing to its maximum value only for CMOD values between 3 ÷ 4 mm

(4)   lower resistance to stable crack growth even though brittle fracture does not occur

Author Response

Respected,

expresses gratitude to the reviewers for well-founded suggestions that contribute to improving the level of quality of work and do not leave the reader with the possibility of doubts and ambiguities. Thank you!

Reviewer

Title: It is suggested to revise the title of the manuscript to “Toughness properties of 50-year-old pipeline material”

Author: Rewiever's remark accepted. Revised the title of the manuscript "Determining the critical toughness value of pipeline material that has been in operation for 50 years" to "Toughness properties of 50-year-old pipeline material"

Abstract:

(1)   Number of sampling should be provided in the abstract

(2)   Significance of the study should be added at the end of the abstract

Author: Rewiever's remarks are all accepted. The abstract was supplemented with the necessary information.

Introduction

Literature study of the following topic supported with current references should be provided:

(1)   Effect on pipe systems towards the functioning of power plants

(2)   Periodic inspections NDT (Non-Destructive Testing) of different types of pipeline system

(3)   Effect of stress intensity factor, integral or critical crack opening on mechanical properties of pipeline

Author: Rewiever's remarks are all accepted. Literature sources were supplemented in accordance with the auditor's remarks. For suggestion No. 1, these are references [1-5]; for suggestion No. 2, these are references [6-8]; for suggestion No. 3 these are references [9-12]. The References part, is aligned with the new references.

Results and discussion

Please justify the following discussion with support of recent references:

(1)   yield strength of the material in the deformed and aged state (state B) is significantly  higher than the value of the yield point of the material in the delivered normalized state of the material

(2)   crack length with the smallest possible plastic zone at the microstructural level.

(3)   crack and plastic deformation with force increasing to its maximum value only for CMOD values between 3 ÷ 4 mm

(4)   lower resistance to stable crack growth even though brittle fracture does not occur

Author: Rewiever's remarks are all accepted.

Suggestion No. 1 – line 71; this statement is based on the numerical values of the mechanical properties of the material in the appropriate state obtained experimentally. In the text, "(Table 1 and Figure 1)" was added to make it clear to the reader on the basis of which parameters the author states that „ ... the value of the yield strength of the material in the deformed and aged state (state B) is significantly higher than the value of the yield point of the material in the delivered normalized state of the material (state A) …”

Suggestion No. 2 – line 100; this statement is supported by new references [13,15,16]

Suggestion No. 3 – line 121; this statement is based on experimentally obtained results by analaysis of the diagram in Figure 5.

Suggestion No. 4 – line 12; this statement is supported by a new reference [17]

Reviewer 3 Report

Review for the

Manuscript ID: sustainability-2240888
Type of manuscript: Article
Title:

Determining the critical toughness value of pipeline material that
have been in operation for 50 years
Author: Darko Bajić *
Submitted to section: Energy Sustainability

   The manuscript analyses the material resistance of NIOVAL47 steel by applying the method for determining R-curves for its three states: normalized state, aged which is 10% cold deformed and heated for 30 minutes at +250°C and deformed state which is 10% cold deformed. The tests show that it is necessary to analyze in detail the results of the aged and deformed state of the material to obtain a reliable picture of the structural integrity of the NO2500 mm pipeline.

   The manuscript is interesting, dedicated for the actual topic, well written and clear and only needs some minor technical corrections and improvements (detailed list in attached PDF file).

After all listed minor corrections will be done, the current manuscript can be accepted for publication.

Author Response

Respected,

expresses gratitude to the reviewers for well-founded suggestions that contribute to improving the level of quality of work and do not leave the reader with the possibility of doubts and ambiguities. Thank you!

Reviewer

page 7, line 189, please correct font for branch junction ( use the same font as for pipeline )

Author: Rewiever's remark accepted. The font size has been corrected.

page 12, Figure 17, please improve technical quality of this illustration by shifting value of the third parameter to the left, because it is overshadowed by the line; parametri su pomjereni

Author: Rewiever's remark accepted. Figure 17 has been corrected.

Something more corrections are requirred for the References part, pages 15 and 16:

For the improvements and corrections please look at: https://www.mdpi.com/authors/references

Cited journals should be abbreviated according to ISO 4 rules, see the ISSN Center's List of Title

Word Abbreviations.

Author: Rewiever's remark accepted. The References part was corrected in accordance with the auditor's recommendation.